# The Effect of Controlled Tile Drainage on Growth and Grain Yield of Spring Barley as Detected by UAV Images, Yield Map and Soil Moisture Content

Renata Duffková [1,*], Lucie Poláková [1,2], Vojtěch Lukas [3] and Petr Fučík [1]

1 Department of Hydrology and Water Conservation, Research Institute for Soil and Water Conservation, Žabovřeská 250, Praha 5, 156 27 Zbraslav, Czech Republic
2 Department of Water Resources and Environmental Modeling, Czech University of Life Sciences Prague, Kamýcká 129, 165 00 Suchdol, Czech Republic
3 Department of Agrosystems and Bioclimatology, Mendel University in Brno, Zemědělská 1, 613 00 Brno, Czech Republic
* Correspondence: duffkova.renata@vumop.cz

**Abstract:** Controlled tile drainage (CTD) practices are a promising tool for improving water balance, water quality and increasing crop yield by raising shallow groundwater level and capillary rise due to drainage flow retardation. We tested the effect of CTD on growth and grain yield of spring barley, at a study site in central Bohemia using vegetation indices from unmanned aerial vehicle (UAV) imagery and Sentinel-2 satellite imagery. Tile drainage flow was slowed by fixed water level control structures that increased soil moisture in the surrounding area according to the terrain slope. Vegetation indices based on red-edge spectral bands in combination with near-infrared and red bands were selected, of which the Normalized Red Edge-Red Index (NRERI) showed the closest relationships with shoot biomass parameters (dry biomass, nitrogen concentration and uptake, nitrogen nutrition index) from point sampling at the tillering stage. The CTD sites showed significantly more biomass using NRERI compared to free tile drainage (FTD) sites. In contrast, in the period prior to the implementation of CTD practices, Sentinel-2 satellite imagery did not demonstrate higher biomass based on NRERI at CTD sites compared to FTD sites. The grain yields of spring barley as determined from the yield map also increased due to CTD (by 0.3 t/ha, i.e., by 4%). The positive impact of CTD on biomass development and grain yield of spring barley was confirmed by the increase in soil moisture at depths of 20, 40 and 60 cm compared to FTD. The largest increase in soil water content of 3.5 vol% due to CTD occurred at the depth of 40 cm, which also had a higher degree of saturation of available water capacity and the occurrence of crop water stress was delayed by 14 days compared to FTD.

**Keywords:** controlled tile drainage; UAV images; red-edge vegetation indices; spring barley biomass; grain yield; soil moisture

## 1. Introduction

In the context of more frequent periods of drought under climate change, it is desirable to reduce water runoff from agricultural land [1–4]. Fields systematically tile-drained to provide suitable conditions for crop production removing excess water from soil profile usually drain water all year round, depending on topography, the crop grown and the hydro-pedological and meteorological conditions [5–8]. Also, nutrients such as nitrogen (N) in the form of nitrates are transported by subsurface drainage at a time when they could be used by growing crops [9–11]. To avoid this inefficient water and nutrients runoff from drained fields, controlled tile drainage (CTD) practices, where drainage flow is retarded by water level control structures (WLCS), are applied especially in some parts of the North America and Europe to retain water and nutrients in drains and related surrounding soil for crop use [12–19]. The purpose of CTD is to increase soil moisture by capillary rise of

the regulated drainage water in the surrounding area, being determined by terrain slope (the so-called range of drainage flow control, RDFC). Drainage water under CTD can only drain from the WLCS when its level reaches the height of the gate (board), inserted in WLCS, falls over it and flows downwards in drainage manholes or outlets. Thus, CTD is a promising approach for improving water balance, water quality and increasing crop yields [5,12,20,21]. The agronomic and environmental benefits of CTD are associated with an increase in soil moisture during the growing season, i.e., better availability of water and nutrients (especially N) for the crops grown [22,23]. Likewise, the formation of an anoxic environment in water-saturated soil for plants by CTD is reduced by diverting the water as soon as the water level reaches a critical level set by the WLCS [5].

Drainage discharge in CTD approach can be controlled by several WLCS types; basically, with adjustable or fixed WLCS placed either at drainage outlets, in manholes or directly on collective or conductive drains [6,15,19,21]. The adjustable WLCS pose an advantage to readily respond to precipitation, runoff conditions and the actual crop water requirements by setting the height of WLCS at the requested level [14]. The WLCS with fixed height could be placed directly on drains and raise the water level according to terrain slope. In more sloping conditions (up to 5%), more densely placed WLCS are needed as their effect on water level rise is less than in flat areas. To assess the effect of increased nutrient and water availability imposed by CTD practices on crop productivity, vegetation indices as spectral reflectance indicators derived from multispectral remote sensing can be useful to diagnose the plant nutritional status and environmental stress symptoms [5,24,25]. Vegetation indices are mostly based on spectral reflectance in the red (R, 630–690 nm) and near-infrared (NIR, 770–1300 nm) bands of the electromagnetic spectrum [26]. Reflectance in the NIR spectrum is related to the plant cell walls and rises as the amount of biomass increases, while reflectance in the R region related to the amount of chlorophyll only in the upper leaves is very low and after reaching a certain amount of the biomass it remains at a minimum constant level, which leads to saturation phenomenon of vegetation indices [27,28]. The commonly used vegetation index associated with R and NIR wavebands is Normalized Difference Vegetation Index (NDVI, [29]) mostly recommended in monitoring green vegetation cover. However, when the vegetation cover becomes dense, i.e., when the leaf area index (LAI) is higher than three, NDVI tends to be saturated, leading to underestimated biomass yield predictions [30–33]. Saturation effect can be reduced by using vegetation indices based on reflectance in a narrow band of the red-edge region (RE, 700–750 nm, e.g., REIP, NDRE, NRERI and RENDVI), showing sensitive increase corresponding strongly to vegetation chlorophyll content and the plant N uptake [34–36]. Mittermayer et al. [37] reported REIP (Red Edge Inflection Point) as a suitable vegetation index to identify site-specific N uptake, high N surplus as well as N loss potential. Vegetation indices based on green reflectance, e.g., GNDVI, GRDVI or MTVI2, also show higher sensitivity to changes in chlorophyll content [38–40].

To demonstrate the prediction of crop biomass production using vegetation indices, yield maps produced in precision agriculture as a result of combine harvester yield sensing system appear to be a useful tool for this purpose. In some cases, vegetation indices are involved within the filters used for spatial interpolation of yield maps that improve the management of soil variability on the farm [41]. Vegetation indices and yield maps were also used to determine the spatial variability of N uptake and N balances [37].

The aim of this study was to evaluate the effect of CTD on the growth and grain yield of spring barley at a study site in Central Bohemia in the 2021 growing season using vegetation indices from unmanned aerial vehicle (UAV) imagery and a yield map at harvest. We hypothesized that CTD would positively affect spring barley development and grain yield through increased soil moisture compared to free tile drainage (FTD).

## 2. Material and Methods

### 2.1. Study Site

We selected two experimental fields (Za Frajmankou, Pod Hvězdou) at the study site in the Central Bohemia near the village of Hvězda-Malíkovice (50.2247167N, 13.9741050E, Figure 1). The local climate is influenced by the rainfall shadow of the Ore Mountains and according to Quitt [42] is classified as warm with a normal of annual rainfall of 501 mm and air temperature of 8.6 °C. In 2021, annual rainfall was 471 mm (growing season 352 mm), and the average annual temperature was 8.6 °C (growing season 14.5 °C). In the partially tile-drained field Za Frajmankou, we applied CTD practices from autumn 2020 to spring 2021 and examined their impact by measuring soil moisture in the 2021 growing season, when spring barley was grown there. In both fields, two UAV images were taken, and spring barley was sampled from selected points twice during the growing season.

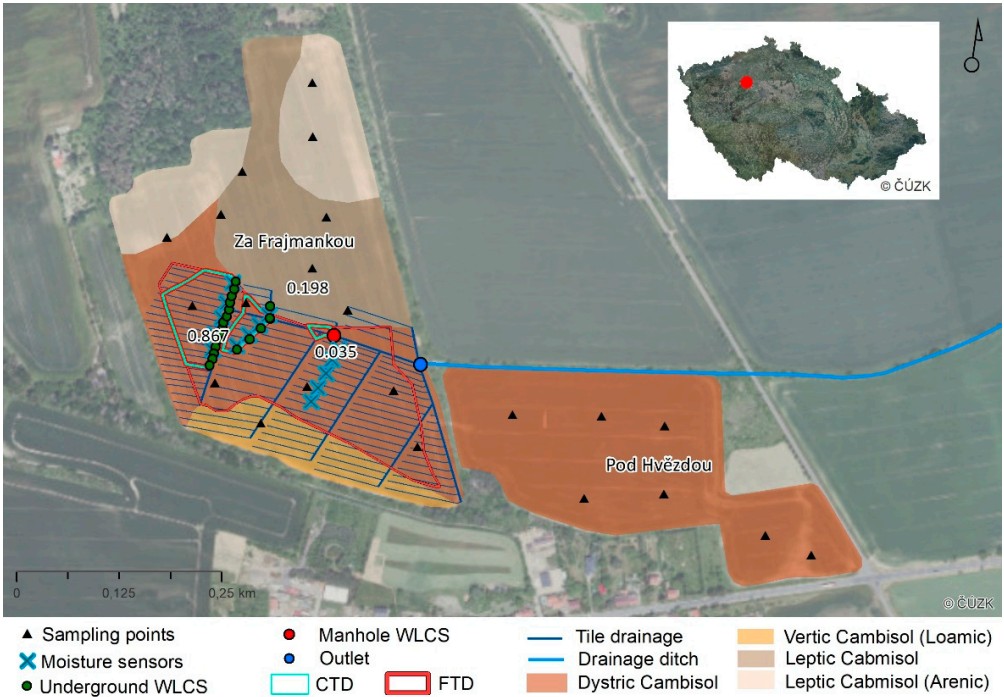

**Figure 1.** Experimental fields Za Frajmankou and Pod Hvězdou with soil types, sampling points, moisture sensors, underground and manhole water level control structures (WLCS), outlet, drainage ditch and map of the Czech Republic with marked study site. CTD—control tile drainage sites, FTD—free tile drainage sites.

The field Za Frajmankou with an area of 14.6 ha, an average altitude of 364.6 m.a.s.l. and a slope of 3° was partially tile-drained in 1962 (7.18 ha), with drainage spacing 10–14 m and lodgement of drains 0.7–1.1 m below surface, drainage water discharging into a drainage ditch (Figure 1). The average slope of the tile-drained area is 3.15° (0.04–9.10°). The field is soil heterogeneous with different types of Cambisols. The lower part of the experimental field with Vertic Cambisol (Loamic) and Dystric Cambisol [43] is texturally diverse (sandy loam, loam, clay loam, clay), predominantly under tile drainage, which developed on sediments from the Permo-Carboniferous period. The higher parts of the field are dominated by sandstone deposits with texturally lighter Leptic Cambisol and Leptic Cambisol (Arenic) classified as sandy loam and loam.

The field Pod Hvězdou (7.4 ha, Dystric Cambisol), located southeast of the field Za Frajmankou, was not drained, and was used only to expand the number of spring barley sampling points to increase the confidence of the statistical analyses.

In both fields, N-P fertiliser (26% N, 14% $P_2O_5$) was applied on 1 April 2021 at a rate of 350 kg/ha (i.e., 91 kg N/ha) and spring barley (variety Solist) was sown on 2 April 2021.

## 2.2. Installation of a Manhole and Water Level Control Structures

The relatively low slope of the field Za Frajmankou provided suitable conditions for slowing drainage flow with CTD practices. In autumn 2020, we installed a total of 20 underground fixed WLCS at 0.7–1.1 m depth below soil surface on selected conductive (main) and collective drains of two drainage groups (Figure 1). The underground fixed WLCS consists of a horizontal PVC pipe with a vertically connected branch (diameter of 110 mm), containing a 6 mm thick polypropylene gate (Figure 2). At all WLCS installation sites, we first removed three original ceramic drainage tiles and replaced them by a PVC drainage pipe with a branch. A gate was inserted into the top hole of the branch and then down leakproof to the PVC pipe. The height of the gate was adjusted so that after inserting the cap at the top of the branch (leaving ca 5 cm for water overflow), approx. 40 cm remained to the soil surface (a safe distance to ensure that the WLCS would not be damaged by ploughing).

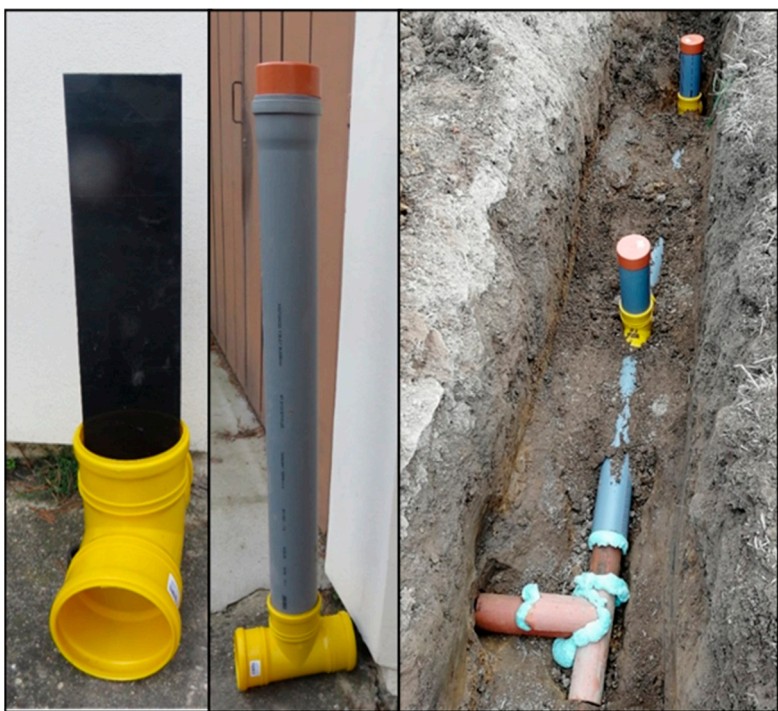

**Figure 2.** Underground fixed WLCS consisting of a drainage T branch (yellow), a polypropylene gate (black) and a PVC pipe vertically connected to the T branch (grey) with a cap (brick-red); before and after installation on tile drainage.

In autumn 2020, we built a control manhole (80 cm in diameter and 1.5 m deep), into which drainage water was connected through 75 mm diameter PVC pipes from CTD and FTD sites and discharged further into the drainage ditch. We installed the same fixed WLCS as on the tile drainage along with a propeller flow meter in the manhole on the PVC pipe bringing water from the CTD sites (Figure 3). This WLCS included a slide valve that, when manually pulled out, would allow the groundwater level (GWL) to be lowered if necessary (as opposed to the underground fixed CWLS). However, it was not needed to control the water level in this way during the monitoring period. The RDFC area of 1.1 ha (0.867 + 0.198 + 0.035, Figure 1) achieved by the installation of both the underground and manhole WLCS and delineated by the known water level (i.e., the height of WLCS) and contours according to DMR 5G (with the declared mean height error 0.18 m in exposed terrain) is considered as an area affected by CTD.

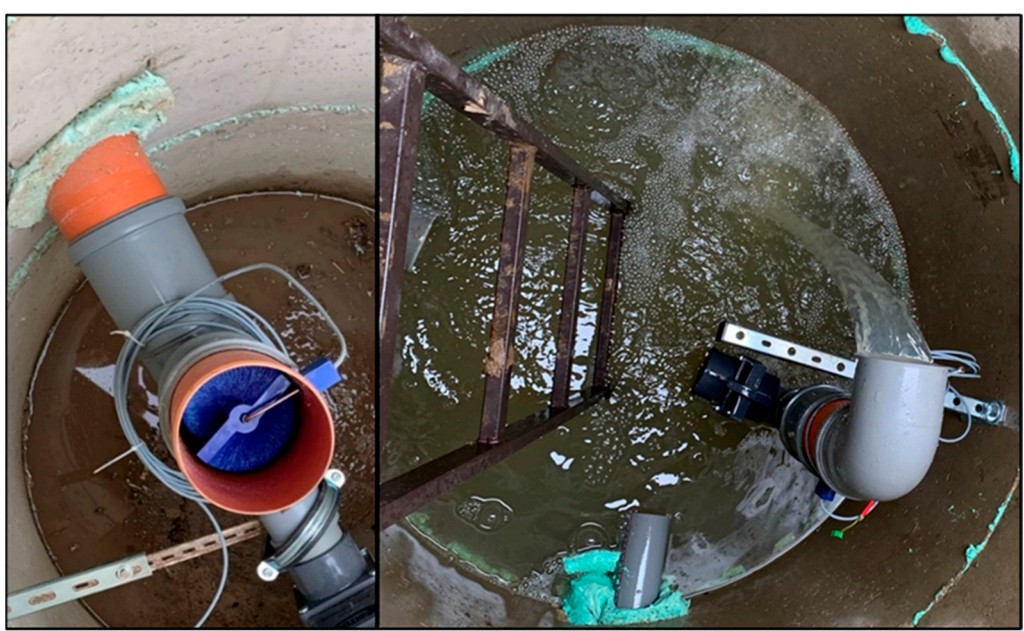

**Figure 3.** Fixed WLCS installed in the manhole with propeller flow meter (left) and water overflow after a rainfall-runoff event on 14 May 2021.

### 2.3. Soil Moisture, Soil Texture and Soil Hydrolimits

To determine the differences in soil moisture between CTD and FTD sites, we measured volumetric water content in the field Za Frajmankou between 19 April and 28 July 2021. We installed a total of 25 sensors (7 sensors at a depth of 20 cm and 9 sensors each at depths of 40 and 60 cm) at 9 locations where RDFC occurred because of increased capillary rise via WLCS (Figure 1). At sites where WLCS were not installed, a total of 15 sensors (5 sensors each at 20, 40, and 60 cm depths) were installed at 5 sites (Figure 1). Soil water volumetric content was measured using the SMT-100 soil moisture probes based on a Time Domain Transmission technology and soil water content reflectometer CS650 (Figure 4). Data were stored in a datalogger at hourly intervals. All sensors were located at Dystric Cambisol.

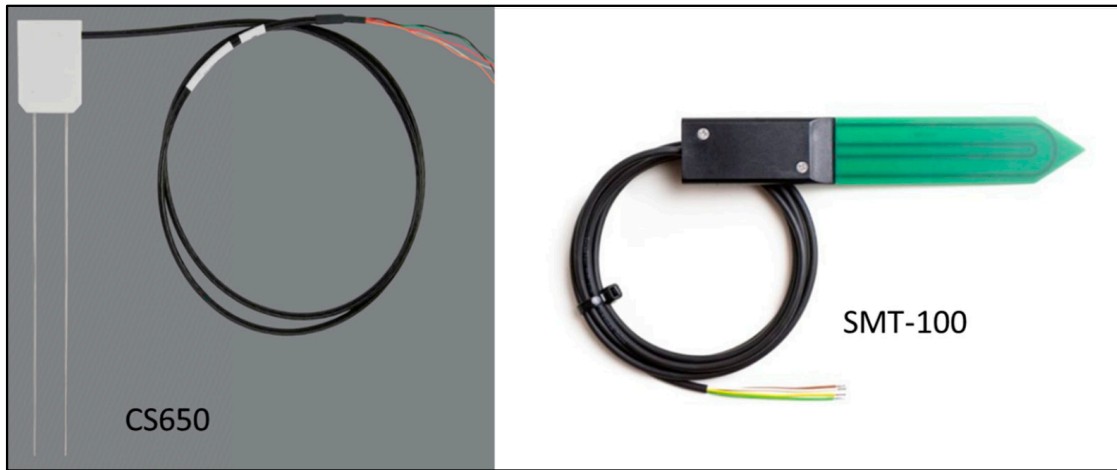

**Figure 4.** Soil volumetric water sensors CS650 and SMT-100.

For each soil moisture sensor, we collected soil samples to determine the soil texture class using the pipette method [44]. A fine particle size fraction (FPSF, %) < 0.01 mm was used to calculate soil hydrolimits: field capacity (FC), point of decreased availability

(PDA), permanent wilting point (WP) and available water capacity (AWC) using simple pedotransfer functions [45]:

$$\text{FC (vol.\%)} = 6.66 + 1.03 \times \text{(FPSF)} - 0.008 \times \text{(FPSF)}^2 \qquad (1)$$

$$\text{WP (vol.\%)} = 2.97 + 0.33 \times \text{(FPSF)} - 0.0012 \times \text{(FPSF)}^2 \qquad (2)$$

$$\text{AWC (vol.\%)} = \text{FC} - \text{WP} \qquad (3)$$

$$\text{PDA (vol.\%)} = \text{WP} + \% \text{ AWC} \qquad (4)$$

FC is the water that remains in the soil after it is thoroughly saturated and free to drain, usually for one to two days. WP is the soil moisture at which plants wilt and do not recover if supplied with sufficient moisture. AWC is the amount of water the soil can retain, and the crops can use. PDA is the minimum soil moisture, expressed as a percentage of AWC, at which plants are still growing and developing successfully (50–60% of AWC for spring barley).

The shallow GWL could have a significant influence on soil water dynamics and is often a profound source of upward water movement by capillary rise. In tile-drained fields across Czechia, based on the experience of the authors from many other sites, the shallow GWL is usually 1.2–1.8 m deep, based on soil morphological and hydrogeological conditions. However, in the trial field, the GWL was not found even at a depth of 2 m in two pits (around the northernmost and southernmost WLCS) which were opened around 3 months prior to WLCS installation. Therefore, we did not measure the GWL depth as it had no effect on soil moisture in the experimental field.

*2.4. Point Sampling and Analysis of Spring Barley Biomass*

We collected spring barley samples from the selected twenty-two points in the Za Frajmankou and Pod Hvězdou fields (Figure 1) at tillering stage (BBCH 25-29) on 2 June 2021 and prior to harvest on 11 August 2021.

Shoot biomass taken on 2 June of 2021 from an area of 0.25 m$^2$ was weighed before and after drying at 105 °C and shoot dry weight was converted to t/ha (dry biomass). Subsequently, the N concentration (%) in shoot dry biomass was determined according to the Kjeldahl method [46]. Shoot N uptake was calculated as dry biomass multiplied by N concentration. To assess plant N status, we calculated nitrogen nutrition index (NNI, [47]) as: $N_{act}/N_{crit}$ where $N_{act}$ is the actual and $N_{crit}$ the critical concentration in dry biomass, respectively. Critical N concentration is the minimum concentrations required to achieve maximum shoot growth and was calculated using the power function capturing a typical dilution curve, i.e., decreasing along with increasing shoot dry biomass. $N_{crit}$ (%) was calculated as 5.35 B$^{-0.442}$ [47], where B is shoot dry biomass (t/ha).

Prior to harvest, on 11 August of 2021, 0.2 m$^2$ of ears were sampled from each sampling point and then, after grain weight was obtained, grain yield (t/ha), grain N concentration according to the Kjeldahl method (%) and grain N uptake (kg/ha) were determined.

*2.5. Vegetation Indices Based on UAV and Satellite Imagery*

Multispectral images for calculation of vegetation indices and assessment of crop status were acquired by UAV imagery near to the date of plant sampling. We conducted UAV surveys on 2 June (BBCH 25-29, tillering stage) and 30 June 2021 (BBCH 51-57, heading stage) by DJI Phantom 4 Multispectral. This UAV is equipped by multispectral camera which capture five narrow spectral bands—blue (B, center wavelength 450 nm), green (G, 560 nm), R (650 nm), RE (730 nm), NIR (840 nm). Simultaneously, the intensity of incoming radiation is recorded by light sensor installed on the upper part of the UAV for the normalization of incoming light conditions. The survey was carried out at a flight altitude of 140 m; based on the sensor resolution of 1600 × 1300 (2.12 MPx) the final spatial resolution provided by images was 7.56 cm. We ensured radiometric calibration of the multispectral camera by scanning the spectral panel Micasense CRP and using

procedures recommended by the manufacturer. Geometric accuracy of acquired images was guaranteed by RTK used in the UAV guidance system and by the placement of 4 ground control points (GCPs) in the observed area.

The orthomosaic of spectral bands was created using the Agisoft Metashape software together with the calculation of the digital surface model (DSM). As a next step, the combined multispectral orthomosaic with all spectral bands was created, from which the set of vegetation indices was subsequently calculated (see Table 1).

**Table 1.** Vegetation indices calculated from the UAV multispectral images.

| | Vegetation Index | Equation | Reference |
|---|---|---|---|
| EVI | Enhanced Vegetation Index | $2.5 \times (NIR - R)/((NIR + 6.0 \times R - 7.5 \times B) + 1.0)$ | [48] |
| EVI2 | Enhanced Vegetation Index 2 | $2.5 \times (NIR - R)/(NIR + 2.5 \times R + 1)$ | [49] |
| GNDVI | Green Normalized Difference Vegetation Index | $(NIR - G)/(NIR + G)$ | [50] |
| SRI | Simple Ratio Index | $NIR/R$ | [50] |
| NDRE | Normalized Difference Red Edge Index | $(NIR - RE)/(NIR + RE)$ | [51] |
| NDVI | Normalized Difference Vegetation Index | $(NIR - R)/(NIR + R)$ | [29] |
| NRERI | Normalized Red Edge Index | $(NIR - RE)/(NIR - R)$ | [33] |
| Chl | Chlorophyll Index | $(NIR - R)/(RE - R)$ | [52] |
| RENDVI | Red-edge NDVI | $(RE - R)/(RE + R)$ | [53] |
| SAVI | Soil Adjusted Vegetation Index | $1.5 \times ((NIR - R)/(NIR + R + 0.5))$ | [54] |

To compare crop development prior to the introduction of CTD practices, we used freely available Sentinel-2 satellite imagery, which we obtained from ESA's free data repositories Openhub, CollGS and Google Earth Engine for the period May to June from 2017–2020. Subsequently, we selected cloud-free images and calculated NRERI values using the formula given in Table 1.

### 2.6. Yield Maps

Crop yield maps were recorded during the harvest of spring barley on 15 August 2021 to analyze the spatial patterns within the field. Raw data were acquired by combine harvester Claas Lexion equipped with sensor system for estimation of grain flow, grain moisture and Differential Global Position System (DGPS) receiver. From the recorded point data, outliers and erroneous values were filtered, followed by spatial interpolation in ESRI ArcGIS using the kriging technique to smooth out the differences at small scale level. The final raster dataset contains information about grain yield in 1 m spatial resolution.

### 2.7. Statistical Analysis

We used linear regression models to determine the relationships between vegetation indices and shoot or grain parameters of spring barley (shoot parameters: dry biomass, N concentration, N uptake, NNI; grain parameters: yield, N concentration, N uptake) taken from twenty-two sampling points in the Za Frajmankou and Pod Hvězdou fields (Figure 1). We included sampling points from the Pod Hvězdou field in models only to provide more data to increase the power of the test. The closest relationship between vegetation indices and shoot parameters, which was derived from a linear logarithmic regression, was exhibited by the Normalized Red Edge-Red Index (NRERI, [33]) based on the RE band. This vegetation index was further used to test the effect of CTD with defined RDFC on the biomass growth and grain yield of spring barley in the Za Frajmankou field. For this, we selected only the predominantly Dystric Cambisol, i.e., 81% of the tile-drained area of the Za Frajmankou field outside the 29 m strip of headlands to exclude the effect of soil type and agricultural machinery movement on crop development.

The NRERI index from the 2 June 2021 UAV image showed a very wide range of unrealistic values ($-476.7$ to $+266.2$) in the $1 \times 1$ m pixels of the selected tile-drained Dystric Cambisol, mainly caused by the movement of agricultural machinery, low vegetation cover or even bare soil. Hence, 6.26% of the pixels were excluded as outliers, mainly in the track lines. No outliers were identified in the UAV image of 30 June 2021. From the Sentinel-2 imagery, from which two images of the Za Frajmankou field (21 May 2017 and 6 May 2018) with grown winter wheat were selected for comparison before the implementation of CTD practices, outliers of the selected tile-drained Dystric Cambisol were identified only in 2018 (9 pixels of $10 \times 10$ m at FTD locations).

Outliers were identified using the 25th (Q1) and 75th (Q3) percentiles and the interquartile range (IQR = Q3 $-$ Q1):

$$\text{Outliers} > Q3 + 1.5 \times IQR \text{ or} < Q1 - 1.5 \times IQR \tag{5}$$

To create a balanced data design, we randomly selected two thousand $1 \times 1$ m pixels with NRERI values from UAV imagery or grain yield values for the CTD and FTD sites using the R script. For Sentinel-2 images, we selected NRERI values from 88 pixels of $10 \times 10$ m from CTD and FTD sites. For normally distributed data (Sentinel-2 from 6 May 2018), an unpaired two-sample *t*-test was used to identify differences in NRERI values between CTD and FTD sites, and in the case of non-normal data distribution detected by the Shapiro-Wilk test (both UAV imagery and Sentinel-2 from 21 May 2017), the Mann-Whitney U test was used as a non-parametric alternative to the independent two-sample *t*-test.

We used Welch's two-sample *t*-test (unequal variances *t*-test) to test for differences in soil moisture measured at CTD and FTD sites. Each data set contained 101 daily averages from all measuring sensors for each depth (20, 40 and 60 cm).

We conducted all statistical analyses in the R environment [55].

### 2.8. Creation of Maps

We analyzed the UAV and Sentinel-2 imagery in ArcGIS software (version 10.7.1). The images were first converted to the S-JTSK coordinate system and then cropped with the required layers to the final image using the Extract by Mask function. The Zonal Statistics and Zonal Statistics as Table functions were used to obtain the mean values. For subsequent statistical analyses in the R environment (*2.7.*), the average pixel values of $1 \times 1$ m (UAV) or $10 \times 10$ m (Sentinel-2) were determined by converting the rasters with the required values to a point layer using the Raster to Point function. We added the required attributes to the resulting layers using the join function. The average NRERI values from the UAVs in the vicinity of the sampling points within a 2-m radius area were obtained using the Buffer and Extract by Mask functions. Outlying points were removed from the point layer of the Sentinel-2 image from 6 May 2018 and the UAV image from 2 June 2021, and the layers were then converted back to a raster layer using the Point to Raster function for the resulting visualization.

### 3. Results

### 3.1. Vegetation Indices

The three vegetation indices from UAV imagery based on reflectance in either the RE, NIR and R regions (NRERI, Canopy chlorophyll content index Chl) or the RE and NIR regions (Normalized Difference Red Edge NDRE) demonstrated the closest relationships with all shoot parameters from point sampling. Of these, NRERI, based on linear logarithmic regressions with the highest adjusted coefficients of determination (R2adj. in Figure 5a–d), was selected as the best indicator of the effect of CTD on growth and grain yield of spring barley. The outlier value of NRERI ($-0.75$) was not excluded as it realistically corresponded to a site with poor stand development.

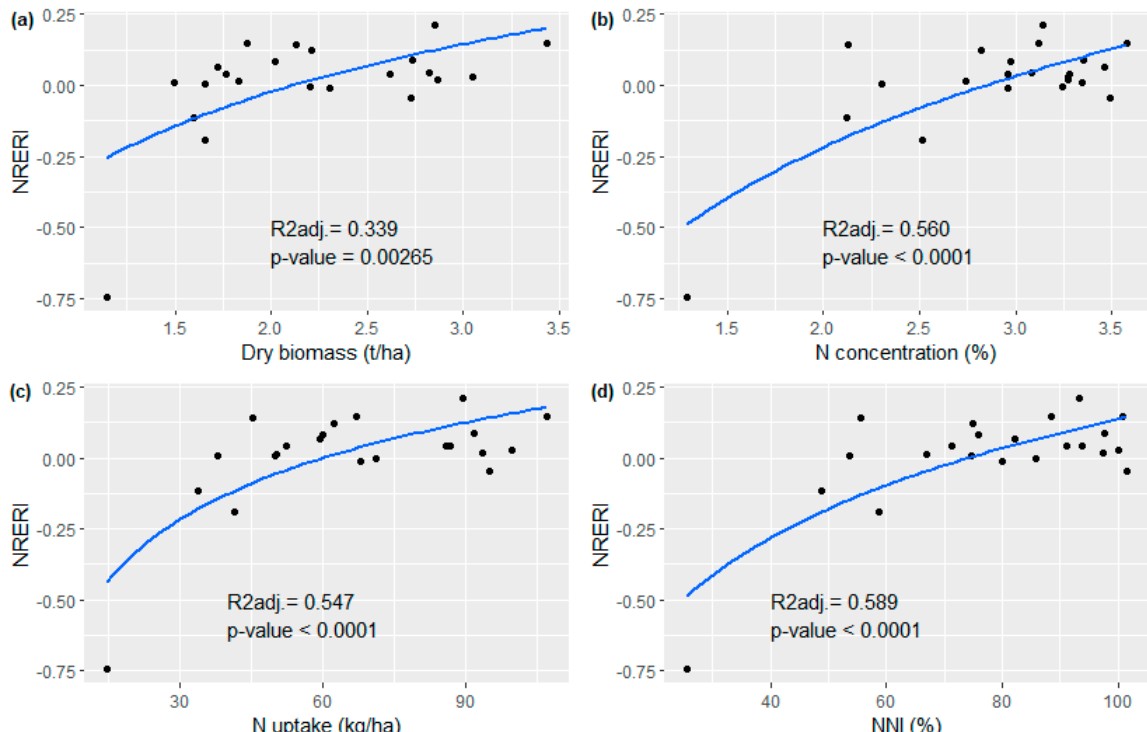

**Figure 5.** Relationships between NRERI from UAV and spring barley shoot biomass parameters ((**a**) biomass dry weight, (**b**) N concentration, (**c**) N uptake, (**d**) NNI) from point sampling of Za Frajmankou and Pod Hvězdou fields on 2 June 2021 expressed by logarithmic functions with ad-justed coefficient of determination (R2adj.) and *p*-value.

NRERI values differed significantly between the two UAV images depending on the different growth stages of spring barley (Table 2). The low, not fully established crop at the tillering stage (2 June 2021) showed a relatively wide range of values even after excluding outliers (−0.20–0.31). The Mann-Whitney U test revealed statistically significant differences between NRERI values in CTD and FTD sites (*p*-value = 0.0006), although not very clear visually (Figure 6a). On the contrary, considerably higher values with a relatively narrow range even without excluding outliers (0.18–0.60) were shown by the fully established stand at the heading stage (30 June 2021) with a height of 50–70 cm. The differences in NRERI values between CTD and FTD sites were more pronounced compared to the previous UAV image (*p*-value < 0.0001, Figure 6b), as also documented in Figure 7 (i.e., higher NRERI in RDFC).

**Table 2.** Means, medians and standard deviations of NRERI from UAV images (without outliers from 2 June 2021) and grain yield from the yield map at controlled (CTD) and free (FTD) sites with Dystric Cambisol in the Za Frajmankou field.

| Date of UAV Image/Harvest | Means | | Medians | | Standard Deviations | |
|---|---|---|---|---|---|---|
| | **CTD** | **FTD** | **CTD** | **FTD** | **CTD** | **FTD** |
| 2 June 2021 (NRERI) | 0.066 | 0.055 | 0.077 | 0.063 | 0.084 | 0.088 |
| 30 June 2021 (NRERI) | 0.434 | 0.395 | 0.435 | 0.395 | 0.032 | 0.033 |
| 15 June 2021 (grain yield, t/ha) | 7.103 | 6.807 | 7.095 | 6.831 | 0.193 | 0.204 |

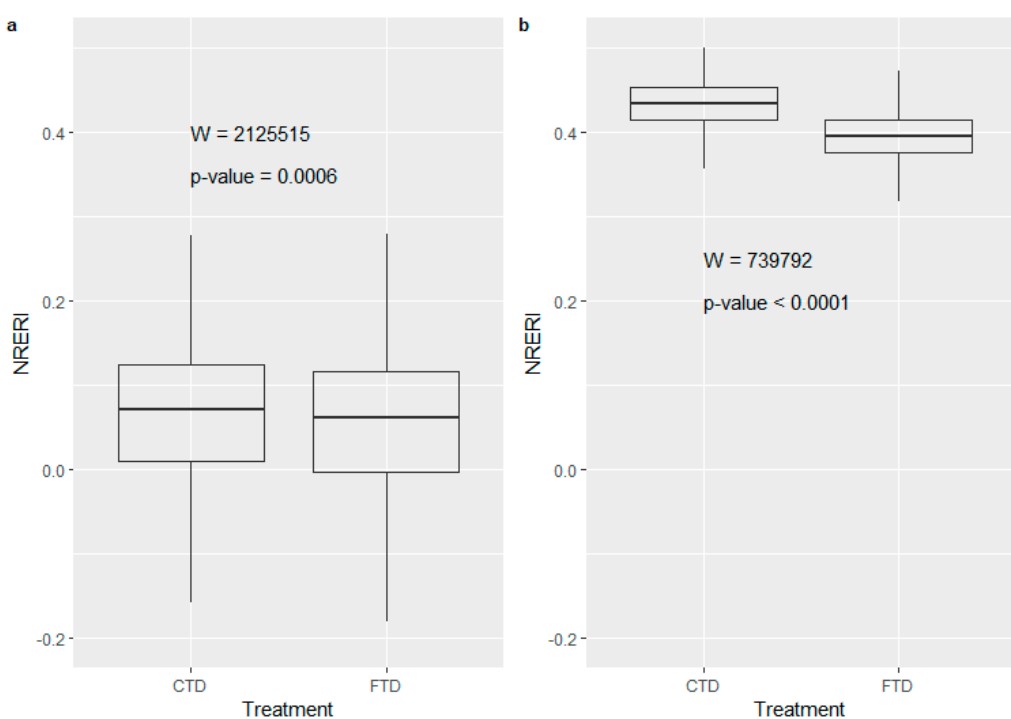

**Figure 6.** NRERI box plots with Mann-Whitney U test (W, *p*-value) at CTD and FTD sites with Dystric Cambisol in the Za Frajmankou field from UAV imagery on (**a**) 2 June 2021 and (**b**) 30 June 2021.

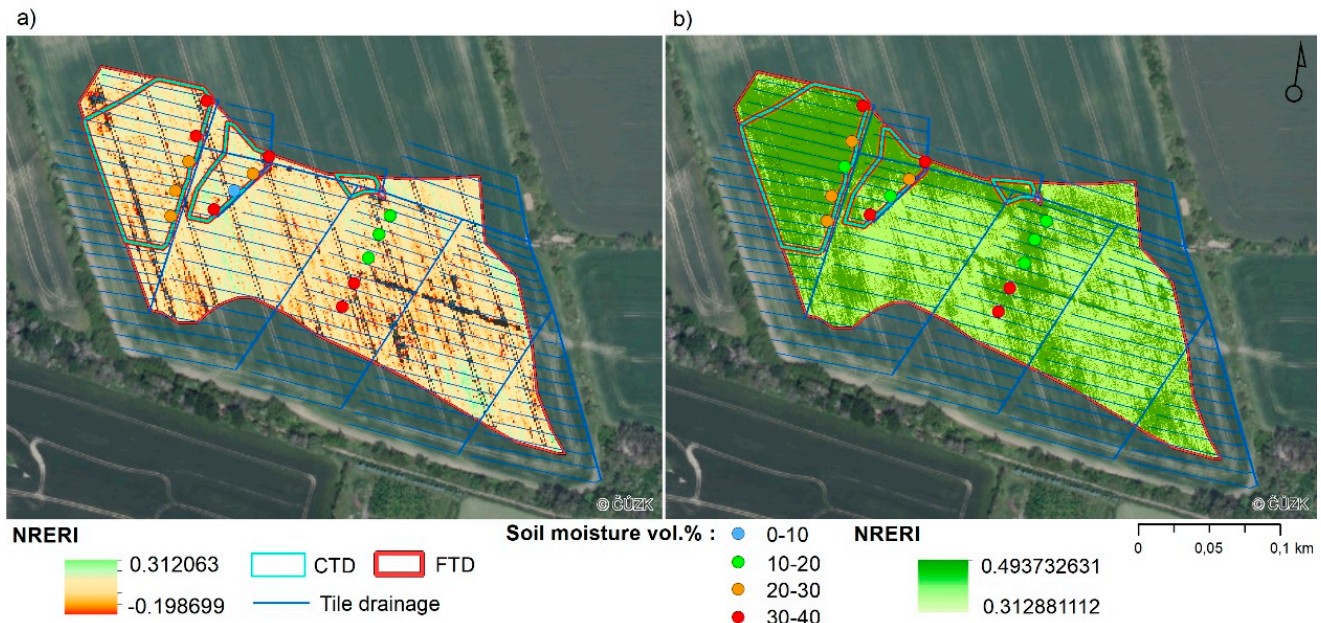

**Figure 7.** Distribution of NRERI and point soil moisture from 40 cm at CTD and FTD sites with Dystric Cambisol (without headlands) of the Za Frajmankou field taken by UAV on (**a**) 2 June 2021 (ex-cluded outliers in black)and (**b**) 30 June 2021.

We found no differences in NRERI values in the two selected Sentinel-2 images between CTD and FTD sites (Figure 8), as demonstrated by the results of the Mann-Whitney U test for 21 May 2017 (W 14 808, *p*-value = 0.5567) and the unpaired two-sample *t*-test for 6 May 2018 (t = −11,834, *p*-value = 0.2383).

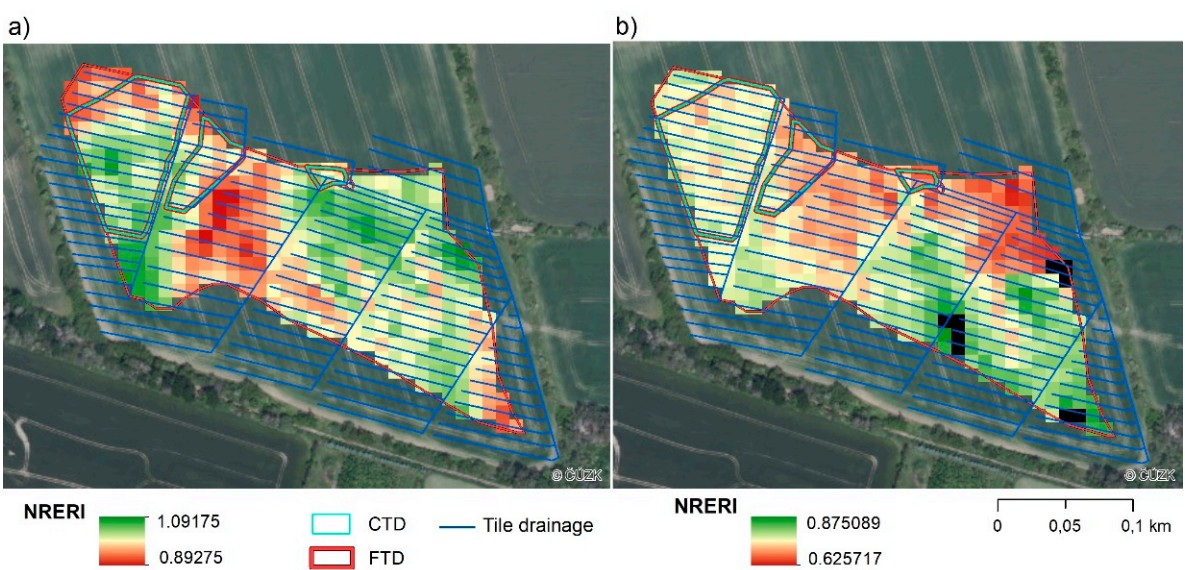

**Figure 8.** Distribution of NRERI at CTD and FTD sites with Dystric Cambisol (without headlands) of the Za Frajmankou field taken from Sentinel-2 on (**a**) 21 May 2017 and (**b**) 6 May 2018 (excluded outliers in black).

*3.2. Grain Yield*

Spring barley grain yield as determined from the yield map taken at harvest on 15 August 2021 was, like the NRERI, significantly affected by CTD (Figure 9). Table 2 shows that grain yield at CTD sites was on average 0.3 t/ha higher than in FTD sites, which was due to the largest and most fertile RDFC area in the western part of the field (Figure 10).

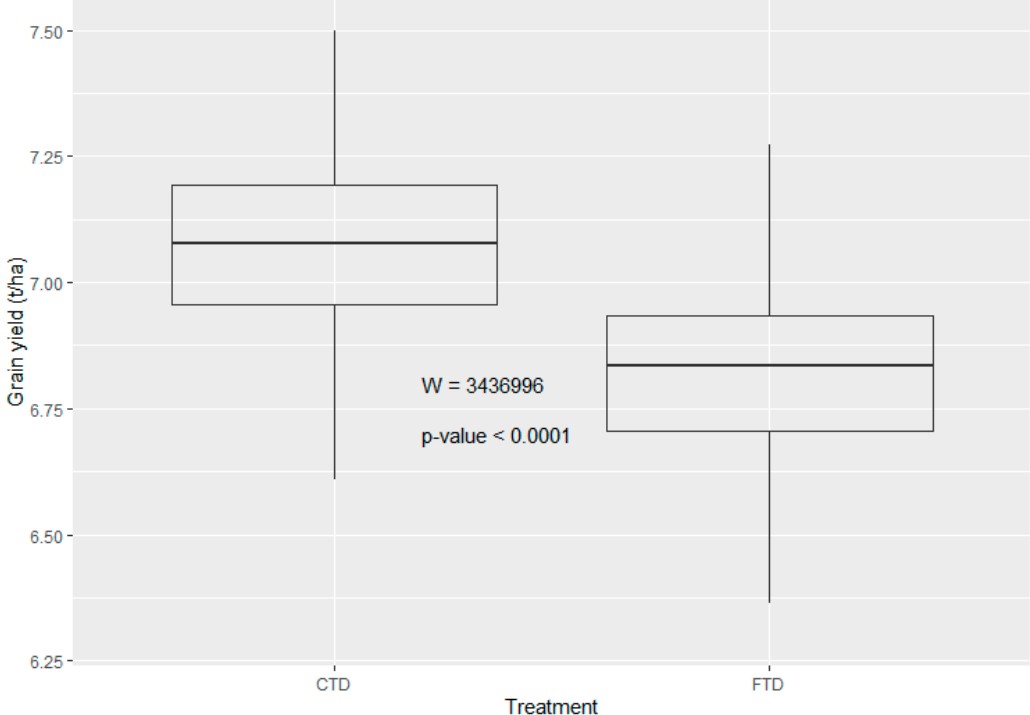

**Figure 9.** Box plots of grain yield with Mann-Whitney U test (W, *p*-value) at CTD and FTD sites with Dystric Cambisol in the Za Frajmankou field from the yield map taken on 15 August 2021.

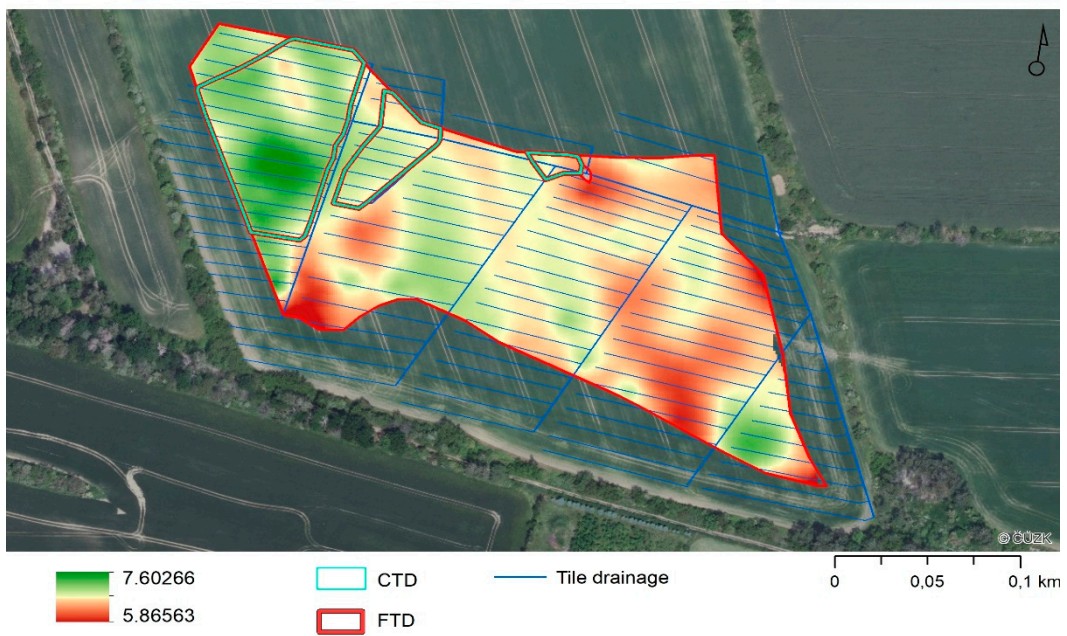

**Figure 10.** Grain yield map of spring barley (t/ha) at CTD and FTD sites with Dystric Cambisol taken at harvest on 15 August 2021.

Correlation analysis was chosen to investigate the relationship between grain yield/grain N uptake and three selected vegetation indices from the sampling points. At the tillering stage (2 June 2021), grain yield and grain N uptake were not correlated with the three selected vegetation indices (NRERI, Chl, NDRE, Table 3). As stand development progressed (30 June 2021), vegetation indices showed statistically significant results in relation to grain yield. The accuracy of estimating grain N uptake using vegetation indices was also improved, but the significance of the relationship was not demonstrated.

**Table 3.** Pearson's correlation coefficients between grain yield and vegetation indices (NRERI, Chl and NDRE) from sampling points on 2 and 30 June 2021.

| Biomass Parameter | Grain Yield | | | Grain Nitrogen Uptake | | |
|---|---|---|---|---|---|---|
| Date | NRERI | Chl | NDRE | NRERI | Chl | NDRE |
| 2.6.2021 | 0.22 | 0.25 | 0.26 | 0.21 | 0.36 | 0.32 |
| 30.6.2021 | 0.51 * | 0.50 * | 0.46 * | 0.40 | 0.41 | 0.36 |

* Significance *p*-value < 0.05.

Correlations of the spatial distribution of NRERI and the yield map on tile-drained Dystric Cambisol without headlands provided similar results (no correlation on 2 June 2021 and weak but statistically significant correlation on 30 June 2021, r = 0.37, *p* < 0.0001).

### 3.3. Soil Moisture Content

Soil moisture values at all depths were significantly higher at CTD sites compared to FTD sites (Table 4, Figure 11). At 40 cm depth, differences were evident throughout the whole study period, thus also at the time of UAV imagery, as shown in Figure 7 in the detail of the fourteen measurement locations. At this depth at the CTD sites, the average amount of water in AWC (AWC saturation level in Table 5) was 10.4% higher, and soil moisture exceeding the soil hydrolimit PDA persisted 14 days longer indicating a delayed onset of crop water stress compared to the FTD sites (Table 5). At other depths, soil moisture exceeding PDA was 4% (20 cm) and 11% (60 cm) higher at CTD sites, but of shorter durations of occurrence.

**Table 4.** Mean soil moisture from each and all depths and Welch´s test parameters (*t*-test, degrees of freedom df, *p*-value) at CTD and FTD sites.

| Depths of Soil Moisture Sensors (cm) | Soil Moisture (vol.%) at Sites: | | *t*-Test | df | *p*-Value |
|---|---|---|---|---|---|
| | CTD | FTD | | | |
| 20 | 22.07 | 20.39 | −2.7307 | 183.13 | 0.0069 |
| 40 | 25.84 | 22.41 | −6.8521 | 179.64 | <0.0001 |
| 60 | 26.07 | 24.25 | −3.0875 | 146.92 | 0.0024 |
| Means of all | 24.66 | 22.35 | −4.3867 | 166.71 | <0.0001 |

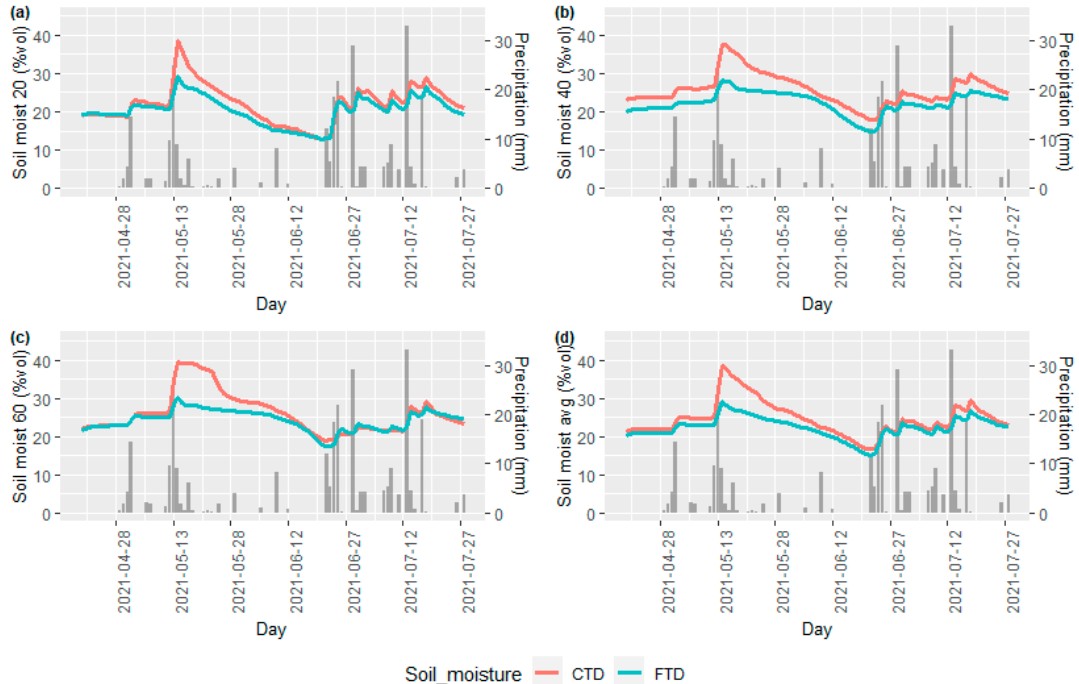

**Figure 11.** Daily precipitation and soil moisture at depths of (**a**) 20, (**b**) 40 and (**c**) 60 cm and (**d**) averaged over all depths at CTD and FTD sites.

**Table 5.** Mean soil hydrolimits (field capacity FC, point of decreased availability PDA, wilting point WP, and available water capacity AWC), degree of saturation of AWC and the number of days when soil moisture was equal to or greater than PDA and FC from each and all depths at CTD and FTD sites.

| Sites | Depth (cm) | FC (vol.%) | PDA (vol.%) | WP (vol.%) | AWC (vol.%) | AWC Saturation Level (%) | Number of Days with Soil Moisture ≥ | |
|---|---|---|---|---|---|---|---|---|
| | | | | | | | PDA (% over PDA) | FC |
| CTD | 20 | 32.6 | 23.4 | 13.0 | 19.6 | 46.4 | 39 (14.3) | 3 |
| | 40 | 34.3 | 25.0 | 14.5 | 19.7 | 57.3 | 52 (16.0) | 6 |
| | 60 | 34.1 | 25.6 | 15.9 | 18.2 | 55.8 | 46 (20.9) | 12 |
| | Means of all | 33.7 | 24.7 | 14.5 | 19.2 | 53.2 | 46 (17.1) | 7 |
| FTD | 20 | 30.0 | 21.2 | 11.7 | 18.3 | 47.4 | 46 (10.4) | 0 |
| | 40 | 31.9 | 23.5 | 14.0 | 18.0 | 46.9 | 38 (7.4) | 0 |
| | 60 | 32.6 | 24.1 | 14.6 | 18.0 | 53.7 | 53 (9.7) | 0 |
| | Means of all | 31.5 | 23.0 | 13.4 | 18.1 | 49.3 | 46 (9.3) | 0 |

The largest increase in soil moisture content, along with its differences at all depths between CTD and FTD sites, occurred after the heavy rains in early May 2021 (Figure 11). Soil moisture was maintained above PDA until the end of May (at 20 cm) or the end of the first decade of June (at 40 and 60 cm). After a rainfall of 47 mm during 11–15 May, soil moisture exceeded even FC values at CTD sites, which remained at 60 cm until May

24, 2021. During this period, water flowed into the manhole through the drainage pipes and flow rates of up to 1.5 l/s were recorded through the WLCS installed in the manhole (Figure 3). This was the only case in the study period where water flowed through the WLCS in the manhole. At the end of May, crop water stress (soil moisture dropped below PDA) occurred at 20 cm depth, which extended to 40 and 60 cm depths during the first decade of June. Soil moisture differences between the CTD and FTD sites at 20 and 60 cm depth gradually decreased until they disappeared completely by the end of the second decade of June. The wet period from 22 June to 28 July 2021 (total rainfall of 181 mm) induced a renewed increase in soil moisture differences at 20 and 40 cm depth and a gradual removal of crop water stress.

## 4. Discussion

Vegetation indices are useful tools of remote sensing for identifying trends in crop biomass growth and predicting crop yields [5]. To test the effect of CTD functionality on biomass production of spring barley in our study, we selected the vegetation index NRERI based on spectral reflectance in R, RE and NIR bands as the best indicator of shoot and grain production. The superiority of the NRERI index (similarly, the NRERI and Chl indices), unlike the other indices based on other combinations of spectral bands (NDVI, SAVI, EVI2, SRI, RENDVI, GNDVI), was due to the better relationships in the case of 2 June 2021 with all shoot biomass parameters and in the case of 30 June 2021 with grain yield from point sampling. The advantage of reflectance in the RE band, in contrast to the R band, is that the sensitivity of absorption to chlorophyll content is much higher (i.e., no saturation effect, [56]) and, similarly to the NIR band, a positive correlation with leaf N and biomass exists [57–60]. The combination of RE and NIR bands is recommended for estimating higher biomass with LAI > 2–3, but also to provide insight into N nutritional status (N content, N uptake, NNI) [61–64]. Thus, the NRERI vegetation index provided an opportunity in our study to show the link between biomass development and the N nutritional status of the crop as well as the prediction of grain yield. Similarly, Klem et al. [65] found that NRERI, as affected by water deficit, is the best estimator of N status in both leaves and grain of winter wheat. Klem et al. [66] also confirmed the suitability of RE reflectance for estimating the N nutritional status of malting barley, the accuracy of which can be further improved by using an artificial neural network based on multiple spectral reflectance wavelengths. Holub et al. [67] reported that at the completed heading stage of winter wheat, the NRERI index, as the only one based on reflectance in the R, RE and NIR spectra, had the greatest potential for estimating grain N uptake.

CTD practices increase crop yields by improving soil moisture availability along with the retention of mineral N available to plants [5,14,68,69]. The beneficial effect of CTD practices on soil water availability for spring barley was clearly reflected in a higher degree of saturation of AWC compared to FTD sites. Also, Wesström et al. [70] found increased soil water storage due to CTD in southern Sweden, which they attributed to reduced drainage outflow compared to FTD. Spring barley has a weaker root system than other cereal crops, and most of its roots (ca. 90%) are distributed at a depth of 30–50 cm depending on soil type, with the highest density at depths up to 10 cm [71–73]. Increased soil water supply at 40 cm in relation to CTD practices, which in our case was maintained throughout the study period, demonstrated an improved water supply to barley roots even when water did not flow through the WLCS in the manhole but was only retained in the drainage pipes. This was highly desirable and clearly contributed to increased grain yield.

The probability of retaining plant-available soil water due to CTD is lower in our drier study site than in humid sites, but even a small increase in soil water availability associated with the elimination of drainage runoff is important for stabilizing or slightly increasing crop yields. For instance, Dou et al. [74] identified a delay in groundwater table decline through CTD (drainage depth of 40 cm) in a dryland area, which allowed sunflower to use groundwater at later growth stages, resulting in yield and water use

efficiency improvements of 4.52–11.14% and 1.16–10.8%, respectively. Accordingly, the grain yield of spring barley in our study was increased by 4% in relation to CTD.

The prediction of grain yield from the early shoot biomass parameters (2 June 2021) from sampling points was not demonstrated, which contrasts with Křen et al. [75] who estimated grain yield of spring barley based on a strong correlation with dry weight of above-ground biomass per unit area at the early growth stage BBCH 25 (r = 0.81). Similarly, the use of selected vegetation indices from the early growth stage for estimating grain yield was not beneficial in our case. As stand development proceeded to the heading stage, the selected vegetation indices based on RE region predicted grain yield at a significant level. Likewise, Erdle et al. [76] demonstrated a close correlation of RE-based vegetation index REIP at later stages of winter wheat development with grain yield. Consistently, Klem et al. [65] considered NRERI, which was measured at the early milk ripening stage, as the best indicator of grain yield in winter wheat. Also, qualitative parameters of harvested crops can be assessed by UAV survey, as shown on the prediction of nutritional values of silage maize using NDVI and NDRE indices by [77].

This study also confirmed the role of UAV multispectral imaging in the monitoring of crop stand and identification of spatial differences in vegetation parameters. The main advantages of UAV in the comparison to the satellite remote sensing, such as free available Sentinel-2 data, are the ultra-high spatial resolution of acquired multispectral data at the few centimeters scale and high operability of drones, which results in the better timing of the survey independent on the cloud condition. Further research is needed for development of collaborative smart drones for fully automated observations [78].

## 5. Conclusions

The use of vegetation indices from UAV imagery based on a combination of R, RE, and NIR wavebands appears to be a suitable method for determining the effect of CTD on biomass growth and N nutritional status of spring barley, as well as for predicting grain yield. CTD practices have shown a distinctly positive impact on biomass development and increased grain yield, as evidenced by increased soil water storage and delayed crop water stress, especially at 40 cm depth. Although this paper describes the results of a field experiment from only one growing season, the effect of CTD on increased biomass growth was clearly demonstrated by Sentinel-2 imagery from before WLCS installation, when there were no differences between CTD and FTD sites. As showed in many other regions, and now also for the Central Europe, the CTD could be thus considered as a measure with a substantial potential to mitigate or delay crop water stress, enhance crop yields, and reduce the undue water loss from the landscape.

**Author Contributions:** Conceptualization, R.D.; methodology, R.D., L.P., V.L. and P.F.; validation, R.D. and V.L.; formal analysis, R.D. and P.F.; investigation, R.D. and V.L.; resources, R.D. and V.L.; data curation, R.D., L.P. and V.L.; writing—original draft preparation, R.D., L.P., V.L. and P.F.; writing—review and editing, R.D., V.L. and P.F.; visualization, R.D., L.P. and V.L.; supervision and project administration, R.D., V.L. and P.F.; funding acquisition, R.D. and P.F. All authors have read and agreed to the published version of the manuscript.

**Funding:** This research was funded by the Technology Agency of the Czech Republic (project No. SS01020309), by The European Union's Horizon 2020 research and innovation program (project No. 818187) and by the Ministry of Agriculture of the Czech Republic (project No. RO0218).

**Data Availability Statement:** The data used for this work are not publicly available.

**Acknowledgments:** The authors would like to thank Zbyněk Kulhavý and Vlastimil Osoba for excellent professional and technical assistance, David Šádek for installation of water level control structures, Darina Heřmanovská advice on the R environment and Ondřej Holubík for soil classification.

**Conflicts of Interest:** The authors declare no conflict of interest.

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
