# Peer review of "The Effect of Controlled Tile Drainage on Growth and Grain Yield of Spring Barley as Detected by UAV Images, Yield Map and Soil Moisture Content"

_remotesensing, doi:10.3390/rs14194959_

Round 1
Reviewer 1 Report
I founbd your work interesting, well described and well documented. My only complain is that most of the manuscript is written in passive voice which makes it more difficult to understand. I encourage you to go over the document and change it to active voice (don't be afraid of using the term "we").
You will find some specific comments on the attached document.

Reviewer 2 Report
3) Materials and Methods
p. 3, Figure 1.
Comment: The RDFC border is poor visible. I suggest to use red, cyan or light green to better visibility.
p. 4, 2.2. Installation of a manhole and water level control structures
Questions: So, did these control structure keep the water level up to 40 cm below soil surface permanently? Did the control structures give ground water level regulation possibilities in case of too high water level?
4) Results
p. 8, lines 310-312: Of these, NRERI was selected as the best indicator of the effect of CTD on growth and grain yield of spring barley with correlation coefficients showing a moderate to strong positive linear relationship (Figure 5a-d).
Comment: Due to Figure 1, CTD is present only at Za Frajmankou, but graphs presented on Figure 5 show correlation between NRERI and parameters (dry biomass, N...) for samples taken on Za Frajmankou and Pod Hvezdou. To see the real effect of CTD on the growth and grain yield and compare with field without CTD, the results should be splitted to two separate graphs. Beside here is not clear, that samples of Za Frajmankou are only from CTD part but CTD and FTD together. This part of text and Figure 5 should be improved or clear explained at least.
p. 10 and 11: Figures: 7, 8 and 10.
Comment: It would be nice, there to mark CTD and FTD areas separately.
p. 12, line 373: (…) Dystric Cambisol without headlands (…)
Comment: What do you mean “headlands”? Hills, uplands, or so?
p. 12. 3.3. Soil moisture content
Comment: Are you able to make a soil moisture map distribution at 40 cm depth? This kind of map could be very interesting and it will show CTD and FTD spots of soil moisture differences. The soil moisture distribution map should be prepared for certain date - f.e. June 30th - consequently to the other results.
p. 13. Figure 11.
Comment: What about ground water level? By my opinion, ground water level strongly determines soil moisture and the graphs of GWL are necessary.
Reviewer 3 Report
the authors highlighted the effect of controlled tile drainage on growth and grain yield 2 of spring barley as detected by UAV images.
Why do you Experimental with fields Za Frajmankou and Pod Hvězdou with soil types?
Authors should improve related work and should highlight the importance of UAV images over satellite image, such as Survey on collaborative smart drones and internet of things for improving smartness of smart cities, Collaboration of UAV and HetNet for better QoS: a comparative study
line 275 is the equation?
in table 1, different equations are used for different work, but it would be great if you mention what equation you used in your paper
figure 5 resolution is low
Round 2
Reviewer 3 Report
The authors addressed my comments very well and i recommend it for publication